# Prognostic Impact of Acute and Chronic Inflammatory Interleukin Signatures in the Tumor Microenvironment of Early Breast Cancer

**DOI:** 10.3390/ijms252011114

**Published:** 2024-10-16

**Authors:** Anne-Sophie Heimes, Ina Shehaj, Katrin Almstedt, Slavomir Krajnak, Roxana Schwab, Kathrin Stewen, Antje Lebrecht, Walburgis Brenner, Annette Hasenburg, Marcus Schmidt

**Affiliations:** Department of Obstetrics and Gynecology, University Medical Center of Johannes Gutenberg University Mainz, 55131 Mainz, Germany; anne-sophie.heimes@unimedizin-mainz.de (A.-S.H.); ina.shehaj@unimedizin-mainz.de (I.S.); katrin.almstedt@unimedizin-mainz.de (K.A.); slavomir.krajnak@unimedizin-mainz.de (S.K.); roxana.schwab@unimedizin-mainz.de (R.S.); kathrin.stewen@unimedizin-mainz.de (K.S.); antje.lebrecht@unimedizin-mainz.de (A.L.); walburgis.brenner@unimedizin-mainz.de (W.B.); annette.hasenburg@unimedizin-mainz.de (A.H.)

**Keywords:** interleukins, breast cancer, tumor microenvironment, gene expression analyses

## Abstract

Interleukins play dual roles in breast cancer, acting as both promoters and inhibitors of tumorigenesis within the tumor microenvironment, shaped by their inflammatory functions. This study analyzed the subtype-specific prognostic significance of an acute inflammatory versus a chronic inflammatory interleukin signature using microarray-based gene expression analysis. Correlations between these interleukin signatures and immune cell markers (CD8, IgKC, and CD20) and immune checkpoints (PD-1) were also evaluated. This study investigated the prognostic significance of an acute inflammatory IL signature (IL-12, IL-21, and IFN-γ) and a chronic inflammatory IL signature (IL-4, IL-5, IL-10, IL-13, IL-17, and CXCL1) for metastasis-free survival (MFS) using Kaplan–Meier curves and Cox regression analyses in a cohort of 461 patients with early breast cancer. Correlations were analyzed using the Spearman–Rho correlation coefficient. Kaplan–Meier curves revealed that the prognostic significance of the acute inflammatory IL signature was specifically pronounced in the basal-like subtype (*p* = 0.004, Log Rank). This signature retained independent prognostic significance in multivariate Cox regression analysis (HR 0.463, 95% CI 0.290–0.741; *p* = 0.001). A higher expression of the acute inflammatory IL signature was associated with longer MFS. The chronic inflammatory IL signature showed a significant prognostic effect in the whole cohort, with higher expression associated with shorter MFS (*p* = 0.034). Strong correlations were found between the acute inflammatory IL signature and CD8 expression (ρ = 0.391; *p* < 0.001) and between the chronic inflammatory IL signature and PD-1 expression (ρ = 0.627; *p* < 0.001). This study highlights the complex interaction between acute and chronic inflammatory interleukins in breast cancer progression and prognosis. These findings provide insight into the prognostic relevance of interleukin expression patterns in breast cancer and may inform future therapeutic strategies targeting the immune–inflammatory axis.

## 1. Introduction

Breast cancer is a heterogeneous disease characterized by different molecular subtypes, each associated with distinct clinical outcomes and therapeutic responses. The tumor microenvironment (TME) plays a pivotal role in shaping the progression and response to therapy in breast cancer [1]. Recent advancements in cancer treatment, as described by Sonkin et al., have focused on exploiting the role of the immune system in the TME, leading to the development of targeted therapies and immunotherapies that have significantly improved patient outcomes [2]. The TME is a dynamic and complex system composed of cancer cells, stromal cells, extracellular matrix, and infiltrating immune cells. Among these components, tumor-infiltrating lymphocytes (TILs) and cytokines, particularly interleukins (ILs), are critical mediators of immune responses and have significant prognostic and therapeutic implications, particularly in triple-negative breast cancer [3,4]. Interleukins are a group of cytokines that facilitate communication between immune cells and can exhibit both protumorigenic and tumor-suppressive properties [5]. Their dual role in cancer biology involves modulating tumor growth, metastasis, and the immune landscape within the TME [6]. For instance, IFN-γ, secreted by activated lymphocytes such as cytotoxic CD8+ T cells and type I CD4+ T helper cells (Th1), strengthens the Th1-mediated antitumor immune response through a positive feedback loop, activating macrophages and enhancing antigen presentation by dendritic cells [7,8]. High levels of IFN-γ are associated with improved survival in breast cancer patients due to its role in orchestrating effective antitumor immunity, though it can also exhibit protumorigenic roles by promoting immune escape mechanisms [6,9]. Previous research has demonstrated that higher IFN-γ mRNA expression significantly impacts metastasis-free survival (MFS) in the basal-like subtype of breast cancer, with higher IFN-γ expression associated with longer MFS. Furthermore, our studies have shown that an elevated expression of an IFN-γ-related gene signature correlates with favorable outcomes across a cohort of 461 patients with early breast cancer [7]. This suggests a broader protective role for IFN-γ and its associated pathways in enhancing antitumor immunity and improving clinical prognosis.

One of the key cytokines that contribute to this protective role is IL-12, which promotes Th1 responses and enhances the cytotoxic activity of Natural Killer (NK) cells and CD8+ T cells, stimulating the secretion of cytokines such as IFN-γ [5,10]. IL-12 has been shown to inhibit tumor growth in breast cancer models by stimulating antitumor immunity and apoptosis in tumor cells [11]. Similarly, IL-21 enhances the proliferation and function of NK cells and CD8+ T cells, boosting antitumor immune responses and reducing tumor burden in preclinical breast cancer studies [12].

Conversely, certain interleukins predominantly exhibit protumorigenic properties. IL-4, IL-5, and IL-13 are associated with the Th2 immune response, which can create an immunosuppressive environment that promotes tumor progression. IL-4 and IL-13, in particular, support breast cancer cell survival, proliferation, and metastasis. In a preclinical breast cancer model, IL-4 has been shown to activate monocytes and tumor-associated macrophages (TAM), facilitating breast cancer metastasis to the lung [13].

IL-10 is known for its ability to suppress effective antitumor immune responses, thereby enhancing tumor immune evasion and progression [14]. Additionally, IL-17 plays a critical role in chronic inflammation and has been reported to promote angiogenesis and tumor growth in breast cancer [15,16]. CXCL1, a chemokine that recruits neutrophils to the tumor site, is associated with tumor progression and metastasis [1,17]. High CXCL1 expression correlates with advanced disease and poor prognosis in breast cancer.

The present study aims to elucidate the prognostic impact of an acute inflammatory versus a chronic inflammatory interleukin mRNA expression signature within the TME of early breast cancer. Additionally, this study investigates correlations between these interleukin signatures and immune cell markers such as CD8, IgKC, and CD20, as well as the immune checkpoint Programmed Cell Death Protein 1 (PD-1), to provide insights into their role in modulating the TME and influencing clinical outcomes in breast cancer.

## 2. Results

The prognostic significance of both interleukin signatures for metastasis-free survival was investigated in a cohort of 461 patients with early breast cancer and long-term follow-up. The analysis included Kaplan–Meier curves and univariate and multivariate Cox regression analyses to assess these signatures in the whole cohort and across different molecular subtypes of breast cancer.

### 2.1. Acute Inflammatory IL Signature

The acute inflammatory IL signature, comprising IL-12, IL-21, and IFN-γ, demonstrated a significant impact on MFS in the basal-like breast cancer subtype. Kaplan–Meier survival analysis indicated that a higher expression of this signature correlated with longer MFS in patients with basal-like tumors (*p* = 0.004, Log Rank), as shown in Figure 1. In the whole cohort, there was a tendency toward a favorable outcome being associated with higher mRNA expression of the acute inflammatory IL signature (Figure 2). However, this effect did not reach significance in the univariate Cox regression analysis (Table 1).

The prognostic significance of the acute inflammatory IL signature was further confirmed in a multivariate Cox regression analysis, adjusted for clinical–pathological variables such as age, tumor size, lymph node status, tumor grade, and Ki-67. The results showed that this signature retained its independent prognostic significance (HR 0.463, 95% CI 0.290–0.741; *p* = 0.001), as summarized in Table 2. Higher expression levels of IL-12, IL-21, and IFN-γ were associated with better outcomes, especially in patients with basal-like breast cancer.

### 2.2. Chronic Inflammatory IL Signature

In contrast, the chronic inflammatory IL signature, including IL-4, IL-5, IL-10, IL-13, IL-17, and CXCL1, showed a significant prognostic effect across the whole cohort of breast cancer patients. A higher expression of this signature was associated with shorter MFS (*p* = 0.034, Log Rank), as indicated in Figure 3.

This suggests that elevated levels of these cytokines correlate with poorer clinical outcomes, highlighting their role in promoting tumor progression and metastasis. The multivariate Cox regression analysis for the chronic inflammatory IL signature is detailed in Table 3, indicating that this signature did not retain independent significance when adjusted for other clinical variables.

### 2.3. Validation of IL Signatures in Independent Cohorts

To validate our results in a larger, independent cohort, we used publicly available gene expression data with associated tumor characteristics and follow-up data [18,19]. The prognostic significance of both the acute inflammatory and the chronic inflammatory IL signatures was analyzed in relation to MFS. The prognostic significance of both the acute inflammatory IL signature (*p* = 0.0039, Log Rank) and the chronic inflammatory IL signature (*p* = 0.015, Log Rank) was successfully validated in previously published datasets (Appendix A Figure A1 and Figure A2).

### 2.4. Correlation with Immune Markers and Immune Checkpoints

To further elucidate the role of these interleukin signatures within the TME, correlations between the interleukin signatures and immune cell markers (CD8, IgKC, and CD20) as well as immune checkpoints (PD-1) were analyzed using the Spearman–Rho correlation coefficient.

The acute inflammatory IL signature showed a strong and significant correlation with CD8 expression (ρ = 0.391; *p* < 0.001), indicating that higher CD8 expression is associated with increased levels of IL-12, IL-21, and IFN-γ (Figure 4). This suggests that these cytokines enhance the presence and activity of cytotoxic T cells within the TME. Positive correlations were also observed between the acute inflammatory IL signature and B cell-associated markers such as IgKC (ρ = 0.157; *p* < 0.001) and CD20 (ρ = 0.160; *p* < 0.001), implying a role in modulating humoral immune responses in the TME (Appendix A Figure A3 and Figure A4).

Similarly, the chronic inflammatory IL signature showed a significant correlation with PD-1 expression (ρ = 0.627; *p* < 0.001). This strong correlation indicates that higher levels of these cytokines are associated with an increased expression of immune checkpoint molecules, which may contribute to immune evasion mechanisms within the tumor (Figure 5).

Overall, these results highlight the contrasting roles of specific interleukins within the TME in breast cancer prognosis. Interleukins such as IL-12, IL-21, and IFN-γ, which are part of the acute inflammatory tumor-suppressive response, are associated with improved outcomes by enhancing antitumor immunity. In contrast, interleukins like IL-4, IL-5, IL-10, and IL-13, which contribute to the chronic inflammatory protumorigenic response, promote tumor progression and immune evasion through their interactions with the tumor microenvironment.

## 3. Discussion

The present study highlights the complex role of two different interleukin signatures in early breast cancer, emphasizing their prognostic significance and interaction within the tumor microenvironment (TME). The acute inflammatory response, characterized by a Th1 profile, drives tumor-suppressive immune activity in the TME, promoting the destruction of tumor cells. However, tumor cells can undergo immunoediting, allowing them to escape immune surveillance. As inflammation progresses from acute to chronic, a Th2 immune profile emerges, leading to immune evasion and tumor promotion [20]. This transition highlights the different immune dynamics captured by the acute and chronic inflammatory signatures in our study. Furthermore, the findings underscore the dual nature of interleukins in modulating immune responses, tumor progression, and metastasis, offering insights into potential therapeutic targets for improving clinical outcomes.

### 3.1. Acute Inflammatory IL Signature: Protective Role in Basal-like Subtype

The acute inflammatory IL signature, comprising IL-12, IL-21, and IFN-γ, demonstrated a significant prognostic impact, particularly within the basal-like breast cancer subtype. This subtype is known for its aggressive nature and poor prognosis, yet our findings suggest that a robust acute inflammatory immune response may improve metastasis-free survival (MFS) in these patients. The independent prognostic significance of this signature, as revealed by multivariate Cox regression analysis, indicates that these cytokines play a critical role in enhancing antitumor immunity.

These results support previous gene expression analysis, which showed that higher IFN-γ expression was associated with better outcomes (longer MFS), particularly in the basal-like subgroup. In addition, a protective effect of an IFN-γ-associated gene signature was demonstrated in the whole cohort of 461 patients with early breast cancer: a higher expression of this signature, covering the IFN-γ signaling pathway, was associated with a favorable prognosis [7]. Jorgovanovic et al. emphasized that anti-PD-1 therapy can be modulated by the interaction between IL-12 and IFN-γ within the tumor microenvironment [6]. Upon binding to PD-1 on CD8+ T cells, these cells are stimulated to produce IFN-γ, which subsequently activates dendritic cells (DCs). This activation increases the secretion of IL-12, which feeds back to further stimulate CD8+ T cells, enhancing IFN-γ production and boosting their cytotoxic activity. This positive feedback loop strengthens immune responses and improves tumor control, as demonstrated in preclinical models using PD-1 antibodies [21]. IL-12 is well known for promoting Th1 responses and enhancing the cytotoxic activity of NK cells and CD8+ T cells, further driving IFN-γ secretion. This cascade stimulates antitumor immunity and inhibits tumor growth and metastasis, as demonstrated in breast cancer models. Notably, preclinical studies have shown that inhibiting the transcription factor YAP activates CD8+ T cells and boosts IL-12 levels, which collectively reduce tumor growth [22].

Using a preclinical mouse model, Mansurov et al. showed that intravenous administration of a collagen-binding domain fused to IL-12 (CBD-IL-12) had a synergistic effect with immune checkpoint inhibitors (ICPi) in mice with immunologically cold tumors, leading to significant tumor regression [23].

IL-21 is another important cytokine that modulates the immune response within the TME of breast cancer. It is produced by activated CD4+ T cells and influences various immune cells, including NK cells, CD8+ T cells, and B cells [24]. IL-21 increases the proliferation and function of NK and CD8+ T cells, thereby enhancing their cytotoxic effects against tumor cells [25]. Additionally, IL-21 promotes the differentiation of B cells into plasma cells, which can produce antibodies targeting tumor cells. By supporting both cellular and humoral immune responses, IL-21 contributes to a comprehensive antitumor effect within the TME.

Interestingly, Sicking et al. evaluated the association between an acute systemic immune response (elevated serum CRP) and gene expression of immune markers of the TME at the mRNA level: they observed no significant correlations between peripheral blood CRP levels and mRNA expression of key markers associated with inflammation (e.g., IL-6 and IL-8), immune response or proliferation in breast cancer tissue [26]. These findings suggest that although CRP is an important systemic marker of inflammation, it does not necessarily reflect localized immune processes within the TME.

The strong correlation between the acute inflammatory IL signature and CD8+ T cell markers suggests that these cytokines enhance cytotoxic tumor-suppressive immune responses within the TME. These findings align with previous studies showing that high levels of TILs, particularly CD8+ T cells, are associated with better prognosis in breast cancer [27,28,29]. In interaction with the cellular immune response, the humoral immune response also plays a crucial prognostic role within the TME. A previous retrospective immunohistochemical study demonstrated that tumor-infiltrating IgKC- and CD38-positive B-lymphocytes were associated with better prognosis in patients with early triple-negative breast cancer [30]. In this context, another important finding of our retrospective study is the correlation of the acute inflammatory IL signature with B cell-associated markers such as IgKC and CD20, indicating that these interleukins may also modulate humoral immune responses.

### 3.2. Chronic Inflammatory IL Signature Adversely Affected Prognosis

In contrast, the chronic inflammatory IL signature, including IL-4, IL-5, IL-10, IL-13, IL-17, and CXCL1, was associated with shorter MFS across the whole cohort. This finding suggests that elevated levels of these cytokines correlate with poorer clinical outcomes, reflecting their role in promoting tumor progression and metastasis.

IL-4, IL-5, and IL-13, which are related to the Th2 immune response, can create an immunosuppressive environment that favors tumor growth [31,32]. These cytokines support breast cancer cell survival and proliferation, contributing to a more aggressive disease phenotype. Higher levels of IL-4 were associated with tumor cell resistance to pro-apoptotic signaling in an in vitro cancer model [31]. In a clinical study by König et al., which evaluated cytokine levels in the serum of breast cancer patients, it was shown that elevated serum levels of both IL-4 and IL-5 were associated with poorer prognosis [33]. Specifically, IL-4 was related to negative estrogen receptor (ER) status and poor differentiation, while IL-5 was associated with positive nodal status [33].

IL-10 can suppress effective antitumor immune responses, facilitating tumor immune evasion by promoting chronic inflammation and a protumorigenic Th2 response [34]. A retrospective study by Llanes-Fernandez et al. revealed that IL-10 can suppress T cell activity, weakening the immune response against tumors. Moreover, its association with markers of apoptosis indicates that IL-10 may facilitate tumor survival by inhibiting the programmed death of cancer cells, which contributes to increased tumor aggressiveness [35]. IL-17, which is involved in chronic inflammation, promotes angiogenesis and tumor growth, further exacerbating disease severity [36]. In a retrospective study by Popovic et al., a negative association between IL-17 and ER expression was shown, suggesting a link between increased IL-17 levels and a more aggressive tumor biology [37]. The chemokine CXCL1, which is expressed in tumor-associated fibroblasts, is associated with poor prognosis. Zou et al. showed that increased CXCL1 expression in breast cancer stroma was associated with higher tumor grade, recurrence, and decreased survival [38].

The strong correlation between the chronic inflammatory IL signature and PD-1 expression highlights the potential for immune suppression and tumor immune escape. PD-1 is an immune checkpoint molecule that inhibits T cell activity, allowing tumors to evade immune surveillance. The association between the protumorigenic cytokines and increased PD-1 expression suggests a mechanism by which these cytokines contribute to an immunosuppressive TME, promoting tumor progression and metastasis.

### 3.3. Therapeutic Implications and Future Directions

The distinct prognostic roles of acute inflammatory and chronic inflammatory interleukin signatures within the TME of breast cancer highlight the importance of a nuanced approach to immunotherapy. Recent studies have highlighted the complex interactions between cytokines and immune checkpoints in cancer progression and treatment responses. Notably, the work of Sonkin et al. emphasizes the critical role of cytokine pathways in shaping the immune landscape, especially in the context of immune checkpoint inhibitors [2]. Although our study focuses on interleukin signatures, integrating additional biomarkers, such as methylation signatures, could further enhance the predictive accuracy for breast cancer survival as well as early breast cancer detection. For instance, a recent systematic review identified several promising methylation biomarkers, including APC, RASSFI, and FOXA1, which show strong potential for early breast cancer detection in non-invasive assays [39]. Future studies could investigate how these distinct biomarker types interact, providing a more comprehensive view of the tumor’s immune landscape and potential therapeutic targets. Targeting the immune–inflammatory axis represents a promising strategy for modulating the TME and enhancing antitumor immunity.

In this context, CD8+ exhausted T cells play a critical role in tumor immune evasion, particularly in the presence of chronic antigen exposure, which leads to functional exhaustion characterized by the expression of inhibitory receptors such as PD-1 [40]. The chronic inflammatory interleukin signature in our study may correlate with an increase in T cell exhaustion, potentially impairing the efficacy of the antitumor immune response. This interaction is particularly relevant in the context of immunotherapies, such as PD-1/PD-L1 inhibitors, which are designed to reverse T cell exhaustion and regenerate cytotoxic T cell activity. The acute inflammatory interleukin signature, on the other hand, may enhance CD8+ T cell function, thereby improving response to immune checkpoint inhibitors. Recent preclinical studies have highlighted the potential for cytokine-based immunotherapies to synergize with immune checkpoint inhibitors: a preclinical study by Wu et al. on the combination of IL-21 and anti-PD-1 therapy demonstrated that IL-21, produced by tumor-associated T follicular helper cells and exhausted CD4+ T cells, can significantly enhance the antitumor effects of PD-1 blockade [41]. This combination drives CD8+ T cells toward clonal expansion and hyperactivity while promoting the differentiation of dendritic cells and M1 macrophages within the TME. These coordinated immune responses reprogram the immune cellular network, increasing the overall efficacy of immunotherapy in preclinical models.

This finding is particularly relevant to our study, as it underscores how IL-21 within the acute inflammatory interleukin signature could enhance immune responses in breast cancer, especially when combined with ICIs. Another study conducted by Pavicic et al. demonstrated the synergistic effect of combining IL-12-based immunotherapy with dual PD-1/CTLA4 inhibition in a preclinical ovarian cancer mouse model [42]. This combination not only enhanced antitumor immunity but also reversed myeloid cell-induced immunosuppression, a common barrier in solid tumors. Similarly, in breast cancer, the modulation of interleukin pathways could potentially increase the efficacy of immune checkpoint inhibitors, making this combination a promising approach for future research and clinical applications.

Future studies could explore whether targeting IL-21 or IL-12 in combination with immune checkpoint blockade could further improve outcomes for patients.

Our study applied mRNA expression analyses to assess the prognostic significance of interleukin signatures in a collective of 461 patients with breast cancer and long-term follow-up data. In recent years, bioinformatics tools, such as those used by Lin et al. [43], who analyzed TRPM7 across multiple cancer types using TCGA datasets, and Tang et al., who examined cuproptosis-related genes in a pan-cancer context [44], have highlighted the utility of bioinformatic approaches in identifying prognostic markers. These studies support the importance of integrating such tools into cancer biomarker research, strengthening the robustness of survival analyses in large datasets and the role of molecular signatures in cancer prognosis and treatment strategies.

The present study provides valuable insights into the prognostic implications of acute and chronic inflammatory interleukin signatures within the TME of early breast cancer. However, it is essential to acknowledge both the strengths and potential limitations of this work.

One of the main strengths of this study is the large cohort size of 461 patients and the long-term follow-up period, with a median follow-up time of 12.75 years, allowing for a comprehensive assessment of survival outcomes. By integrating established prognostic factors such as tumor grade, size, nodal status, and molecular subtypes, this study enhances the clinical relevance and applicability of the findings. The dual focus on both acute and chronic inflammatory cytokines offers a balanced perspective on the complex role of the immune response in breast cancer, elucidating the contrasting effects of different cytokines on tumor progression and patient outcomes.

Despite these strengths, this study has several limitations. The retrospective design of this study may introduce biases related to patient selection and data collection. As a single-center study, the findings may have limited generalizability to other populations and clinical settings, necessitating multi-center studies for validation. Additionally, it is important to note that over half of the patients in our cohort received adjuvant systemic therapy (either endocrine therapy or chemotherapy). This may impact the strength and interpretation of the interleukin signatures as prognostic factors. While this study provides strong associations between interleukin expression and clinical outcomes, functional validation experiments to elucidate the mechanistic roles of specific interleukins were not performed. Additionally, this study does not extensively analyze the interaction between interleukin expression and specific treatment modalities, which could provide more targeted clinical insights. Therefore, future studies should aim to stratify patients based on their systemic therapy to more clearly understand the prognostic and possibly predictive value of these interleukin signatures.

In summary, while this study significantly contributes to understanding the prognostic role of interleukin signatures in early breast cancer, it also highlights areas for further research and validation. Addressing these limitations in future studies will be crucial for translating these findings into clinical practice and improving patient outcomes.

## 4. Materials and Methods

### 4.1. Patient Cohort

This study cohort included 461 patients with early breast cancer who underwent surgery at the Department of Gynecology and Obstetrics at the University Medical Center Mainz between 1986 and 2000. Adequate tumor tissue (fresh frozen) was available for successful Affymetrix microarray analysis. The cohort comprised three subgroups with different systemic treatments:N0 cohort: 200 node-negative patients who received no further adjuvant therapy after surgery and radiation;Tamoxifen cohort: 165 patients treated with tamoxifen as a single adjuvant therapy;Chemotherapy cohort: 96 patients treated with either cyclophosphamide, methotrexate, fluorouracil (CMF; n = 34) or epirubicin, cyclophosphamide (EC; n = 62) in the adjuvant setting.

### 4.2. mRNA Isolation and Gene Expression Analysis

The mRNA isolation and gene expression analysis were performed as previously described [7,45]. Tumor samples were frozen and stored at −80 °C. Approximately 50 mg of frozen breast tumor tissue was fragmented in liquid nitrogen. RLT buffer was added, and the homogenate was centrifuged through a QIAshredder column (Qiagen). Total RNA was isolated from the eluate using the RNeasy kit (Qiagen) according to the manufacturer’s instructions. RNA yield was determined by UV absorbance, and RNA quality was assessed using an Agilent 2100 Bioanalyzer RNA 6000 LabChip Kit (Agilent Technologies, Santa Clara, CA 95051, USA).

For mRNA expression analysis, fresh frozen tumor tissue from 461 breast cancer samples was used to generate HG-U133A arrays (Affymetrix, Santa Clara, CA, USA) and measure the relative transcript frequencies of the selected genes. Labeled cRNA was synthesized from 5 µg of total RNA using Roche Microarray cDNA Synthesis, Microarray RNA Target Synthesis (T7), and Microarray Target Purification Kits (Roche Applied Science, Mannheim, Germany) following the manufacturer’s instructions. Raw expression data (CEL files) were standardized by multiarray analysis (fRMA). The complete record of 461 samples used in the current study with updated follow-up is filed at the NCBI in the GEO database under accession number GSE158309.

The gene expression data included the following single genes and corresponding probe sets related to acute and chronic inflammatory interleukin signatures:

Chronic inflammatory interleukin signature:
IL-4: 207539_s_at, 207538_at;IL-5: 207952_at;IL-10: 207433_at;IL-13: 207844_at;IL-17: 208402_at, 220273_at, 220971_at;CXCL1: 204470_at.

Acute inflammatory interleukin signature:
IL-12: 207901_at, 207160_at;IL-21: 221271_at;IFN-γ: 210354_at.

An overall interleukin signature was calculated as the average of the expression values of the chronic inflammatory interleukins (IL-4, IL-5, IL-10, IL-13, IL-17, and CXCL1) and acute inflammatory interleukins (IL-12, IL-21, and IFN-γ). Scores above the median of the inflammatory signature were defined as high expression, and scores below the median were defined as low expression.

To validate our results on a larger, independent cohort, we used publicly available gene expression data of both signatures with associated tumor characteristics, as well as follow-up data [18,19].

### 4.3. Molecular Subtypes

Intrinsic subtypes were determined according to the model proposed by Haibe-Kains et al., which included estrogen receptor gene (ESR1), HER2, and Aurora kinase A (AURKA). ER and HER2 status were determined from the bimodally distributed mRNA levels of ESR1 (205225_at) and ERBB2 (216836_s_at), respectively. The cut-off values for ESR1 and ERBB2 were determined using model-based clustering and the upper quartile plus interquartile range of mRNA levels. The median mRNA expression of AURKA (208079_s_at) was used as the cut-off value.

This resulted in the following molecular subtypes:ESR1-positive, HER2-negative, and low proliferation (AURKA low) luminal A-like;ESR1-positive, HER2-negative, and high proliferation (AURKA high) luminal B-like;HER2-positive;ESR1-negative and HER2-negative basal-like.

Table 4 shows the absolute and relative frequencies of these molecular subtypes within the patient cohort.

### 4.4. Statistical Analysis

Statistical analyses were performed using SPSS version 28.0 (SPSS Inc, Chicago, IL, USA) and Stata version 17. The prognostic significance of pro- and anti-inflammatory interleukin expression for metastasis-free survival (MFS) was examined by Kaplan–Meier survival analysis (≤median vs. >median) and univariate and multivariate Cox regression analysis. Multivariate Cox regression analysis was adjusted for tumor stage (T1 vs. T2, T3, and T4), histological grade (GI + GII vs. GIII), Ki-67 (≤20% vs. >20%), and lymph node status (negative vs. positive). The significance of Kaplan–Meier survival analysis was assessed using the log-rank test. Correlations were analyzed using the Spearman–Rho correlation coefficient.

## 5. Conclusions

This study highlights the prognostic relevance of an acute inflammatory versus a chronic inflammatory interleukin mRNA expression signature in early breast cancer. The acute inflammatory signature, particularly in the basal-like subtype, is associated with improved metastasis-free survival, suggesting a protective role in enhancing antitumor immunity. Conversely, the chronic inflammatory IL signature correlates with poorer outcomes, indicating its role in promoting tumor progression and immune evasion. These findings underscore the potential of targeting the immune–inflammatory axis in breast cancer therapy. Future research should focus on validating these results in larger cohorts and exploring the underlying mechanisms to develop targeted therapeutic strategies that improve patient outcomes.

## Figures and Tables

**Figure 1 ijms-25-11114-f001:**
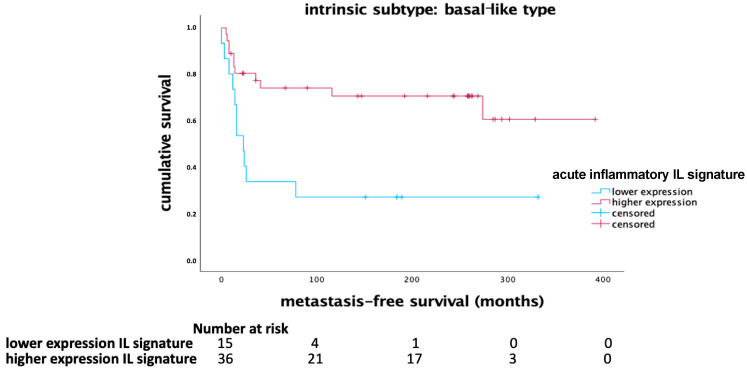
Kaplan–Meier diagram of the acute inflammatory IL signature (IL-12, IL-21, and IFN-γ) in the basal-like subtype (*p* = 0.004, Log Rank).

**Figure 2 ijms-25-11114-f002:**
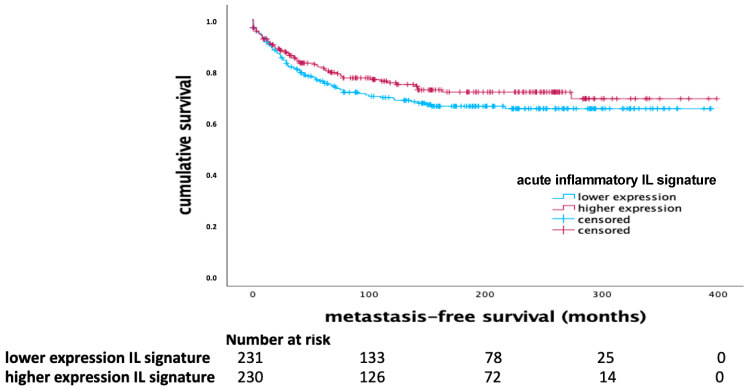
Kaplan–Meier diagram of the acute inflammatory IL signature (IL-12, IL-21, and IFN-γ) in the whole cohort (*p* = 0.205, Log Rank).

**Figure 3 ijms-25-11114-f003:**
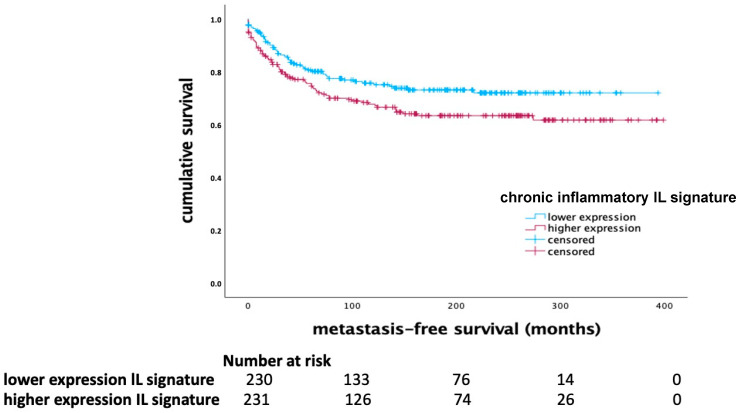
Kaplan–Meier diagram of the chronic inflammatory IL signature (IL-4, IL-5, IL-10, IL-13, IL-17, and CXCL1) in the whole cohort (*p* = 0.034, Log Rank).

**Figure 4 ijms-25-11114-f004:**
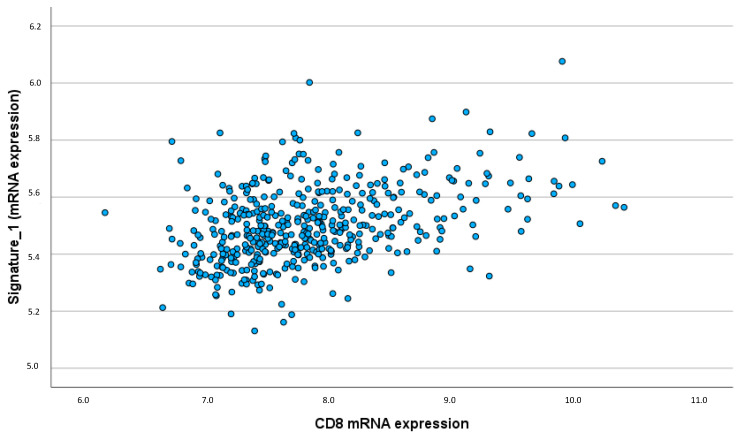
Correlation acute inflammatory IL signature/CD8 expression (ρ = 0.391; *p* < 0.001).

**Figure 5 ijms-25-11114-f005:**
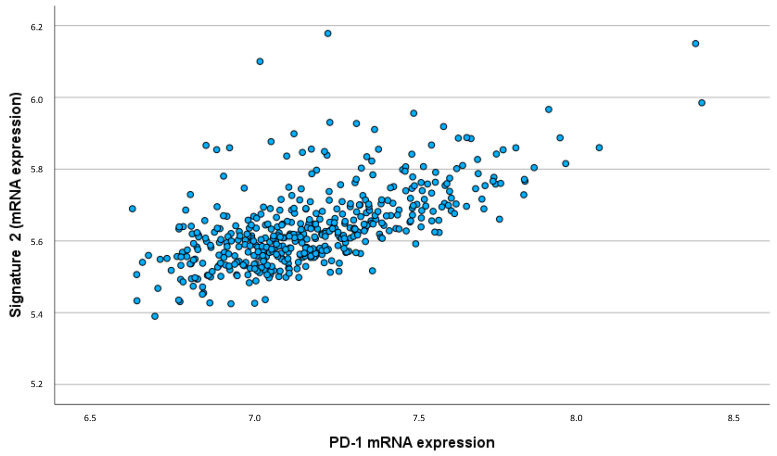
Correlation chronic inflammatory IL signature/PD-1 expression (ρ = 0.627; *p* < 0.001).

**Table 1 ijms-25-11114-t001:** Univariate Cox regression analysis.

		HR	95% CI		*p*-Value
			Lower	Upper	
Chronic inflammatory IL signature	High vs. Low	1.446	1.025	2.041	0.036
Acute inflammatory IL signature	High vs. Low	0.802	0.569	1.129	0.206
Age	<50 vs. ≥50	0.851	0.578	1.253	0.415
Tumor size	T2-4 vs. T1	2.219	1.513	3.254	<0.001
Lymph node status	N1,2,3 vs. N0	2.288	1.598	3.276	<0.001
Grade	GIII vs. GI/II	4.946	2.020	12.109	<0.001
Ki-67	>20% vs. <20%	1.926	1.254	2.957	0.03

**Table 2 ijms-25-11114-t002:** Multivariate Cox regression analysis (acute inflammatory IL signature).

		HR	95% CI		*p*-Value
			Lower	Upper	
Acute inflammatory IL signature	High vs. Low	0.463	0.290	0.741	0.001
Age	<50 vs. ≥50	1.265	0.729	2.195	0.403
Tumor size	T2-4 vs. T1	1.604	0.995	2.585	0.052
Lymph node status	N1,2,3 vs. N0	1.479	0.926	2.365	0.102
Grade	GIII vs. GI/II	2.489	1.527	4.057	<0.001
Ki-67	>20% vs. <20%	1.660	1.025	2.688	0.039

**Table 3 ijms-25-11114-t003:** Multivariate Cox regression analysis (chronic inflammatory IL signature).

		HR	95% CI		*p*-Value
			Lower	Upper	
Chronic inflammatory IL signature	High vs. low	1.098	0.714	1.689	0.671
Age	<50 vs. ≥50	1.242	0.716	2.156	0.440
Tumor size	T2-4 vs. T1	1.510	0.938	2.430	0.090
Lymph node status	N1,2,3 vs. N0	1.207	0.762	1.912	0.422
Grade	GIII vs. GI/II	2.218	1.369	3.595	0.001
Ki-67	>20% vs. <20%	1.516	0.944	2.436	0.085

**Table 4 ijms-25-11114-t004:** Patient’s characteristics.

	**Number of Patients** **(n = 461)**	**Percentage (%)**
Age at diagnosis		
≤50 >50	104 357	22.6 77.4
Tumor size		
T1 T2 T3 T4 Missing value	188 214 19 39 1	40.8 46.4 4.1 8.5 0.2
Tumor grade		
GI GII GIII	63 287 111	13.7 62.3 24
Lymph node status		
N0 N1 N2 Nx	254 140 49 18	55.1 30.4 10.6 3.9
Tumor type		
Invasive ductal (NST) Invasive lobular Others	291 79 91	63.1 17.1 19.7
ER		
Positive Negative Missing value	381 79 1	82.6 17.1 0.2
PR		
Positive Negative Missing value	346 114 1	75.1 24.7 0.2
HER2		
Positive Negative Missing value	46 358 57	10 77.7 12.3
Ki-67		
>20% ≤20% Missing value	138 250 73	29.9 54.2 15.8
Molecular subtypes		
Luminal A-like Luminal B-like Basal-like HER2-positive	189 182 51 39	41 39.5 11.1 8.5
Distant metastasis		
Yes No	133 328	28.9 71.1
Treatment collective		
N0, untreated Endocrine treatment (tamoxifen) Chemotherapy: • CMF • EC	200 165 96: • 34 • 62	43.4 35.8 20.8: • 7.4 • 13.4

## Data Availability

Data is contained within the article.

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
