# Peer review of "Prognostic Impact of Acute and Chronic Inflammatory Interleukin Signatures in the Tumor Microenvironment of Early Breast Cancer"

_ijms, 2024, doi:10.3390/ijms252011114_

Round 1
Reviewer 1 Report
Comments and Suggestions for Authors
The study employs rigorous statistical analyses (Kaplan-Meier curves and Cox regression) and a sizable patient cohort, enhancing the reliability of the findings. The research addresses an essential aspect of breast cancer prognosis by exploring the roles of interleukin signatures, contributing valuable insights into potential therapeutic targets.
However, the paper could benefit from a broader range of references, especially recent studies that explore the role of other cytokines in cancer, to contextualize its findings within the current research landscape. More comparisons with similar studies could strengthen the argument by highlighting where this study's findings align or diverge from existing research. The discussion could delve deeper into the biological mechanisms behind the observed effects of interleukin signatures. Speculating on the pathways and interactions at a cellular level might provide a richer theoretical framework for the results.Expanding on how these findings could influence clinical practice would be beneficial. For instance, discussing potential changes in therapeutic strategies or patient management could make the research more applicable to clinical settings.While the paper briefly touches on future research, detailing specific studies that could be conducted based on these results, such as clinical trials or longitudinal studies, would be advantageous.The study acknowledges some limitations, but a more thorough examination of potential biases and the generalizability of the results would enhance the paper's credibility. Discussing limitations in depth helps in setting the scope and reliability of the findings.Including more about the validation of the interleukin signatures using independent cohorts would solidify the findings. Additionally, discussing any discrepancies in validation results could provide a more balanced view.
Details comments: Cite More Diverse Sources. Incorporating a wider range of sources, including meta-analyses and systematic reviews, could provide a more comprehensive background and validate the significance of the interleukin signatures discussed. In the introduction, You should mention the overview of the cancer and cancer treatment, suggested to refer to PMID: 38909530 by Dr Beverly A. Teicher. The paper should also mention Methylation signatures for breast cancer biomarker (10.1016/j.cancergen.2023.12.003).Integrate a section that reviews and discusses previous bioinformatic studies using similar tools to study cancer biomarkers. PMID: 36332746 by Jun Lin. PMC9582932 used GSCAtool. These are similar studies doing survival analysis. CD8 exasted T cells should be definitely discussed. A detailed discussion on how these interleukin signatures might interact with different treatment modalities (chemotherapy, immunotherapy) could be enlightening, particularly in how they might affect treatment outcomes. Extending the discussion to include broader implications, such as impacts on different patient subgroups or healthcare systems, could make the paper more comprehensive and impactful.
Reviewer 2 Report
Comments and Suggestions for Authors
The manuscript titled Prognostic impact of pro- and anti-inflammatory interleukin mRNA-expression signatures in early breast cancer reports the association of pro- and anti-inflammatory gene expression signature detected in breast cancer tissues with metastasis-free survival of early breast cancer patients.
This is a retrospective study with an interesting approach. However, anti- and pro-inflammatory interleukin signatures include well-known inflammatory mediators that are surprisingly mentioned in the opposite group. For example, in the current study, IFN-g is part of anti-inflammatory interleukin signature while it has been mentioned as a pro-inflammatory cytokine and the fact that “IFN-g can enhance Th1-mediated anti-tumor immune response” by the same authors in Int. J. Mol. Sci. 2020, 21, 7178. IL-10, one of the most potent anti-inflammatory cytokines, is mentioned in the pro-inflammatory signature, which will confuse most readers and experts in the Immunology. Therefore, the definition of both signatures as “anti-inflammatory” and “pro-inflammatory” must be clearly validated based on the immune cell response. furthermore, it is troubling to understand that “higher expression of this anti-inflammatory signature correlated with longer MFS in patients”, which assumes a flawed concept that blocking the anti-tumor immune response would lead to tumor regression while Immunotherapy, including immune checkpoint inhibitor therapy, is about activating the immune response and inhibiting immune evasion for tumor regression. The same misinformation regarding the supposed pro-inflammatory interleukin signature, which contains classical anti-inflammatory mediators, describes an expected outcome that high expression level is associated with poor prognosis due to tumor growth. Knowing that T cells express CD8 and PD-1on their surface, it would have been preferable to select another inflammatory marker more characteristics to pro- and anti-inflammation. Of note, the Table 1 is not mentioned in the text of the Results section and there are no Figures showing the results related to IgKC and CD20. Regarding the writing of the manuscript, there are too many sentences mentioned in the Introduction that are too similar and almost repeated in the Discussion. The Table 4 of Patients’ characteristics present data that differ from the above-mentioned paper, such as those displayed in the Tumor grade category, authors should explain the distribution of data from the 32 missing values mentioned in the previous paper. Same remark for Lymph node status category. Surprisingly, the number corresponding to PR positive/negative tissues and Distant metastasis have changed. This must be clarified as well. In the Conclusions, the authors could only state that “The anti-inflammatory interleukin signature, .., is associated with improved metastasis-free survival, suggesting a protective role in enhancing anti-tumor immunity”, and that “pro-inflammatory interleukin signature correlates with poorer outcomes, indicating its role in promoting tumor progression and immune evasion”, which contradicts the basic concept of pro-inflammatory mediators activating immune cell response for tumor regression. Therefore, due to the numerous flaws concerning the basic concept of Immunology, Immunotherapy, including immune checkpoint inhibitor therapy, and the lack of explanation of the changes in the Patients’ characteristics data, this manuscript as it stands should not be considered for publication.
Comments on the Quality of English LanguageMinor grammatical errors.
Round 2
Reviewer 1 Report
Comments and Suggestions for Authors
It is good now.
Author Response
Reviewer 1:
Reviewer 1:
“It is good now.”
We want to thank the reviewer and value the reviewer's positive assessment in round 2, after we received many useful comments and suggestions in round 1 that improved our manuscript.
Reviewer 2 Report
Comments and Suggestions for Authors
I would like to thank the authors for considering my comments. However, the modifications made including the changes in the Title, raise more questions. By replacing anti-inflammatory and pro-inflammatory with acute and chronic inflammatory, respectively, the authors failed to explain the biological criteria selected to define the two phases of inflammation. Knowing that acute inflammation causes the production of acute phase proteins, the authors must therefore provide the plasma concentration of C-reactive protein detected in the patients, for instance. In addition, IL-6, IL-1b, IL-8, TNF-a and TGF-b are well known as acute inflammatory interleukins, so the authors are strongly recommended to include at least IL-6 and IL-8 to the acute inflammatory IL signature.
Comments on the Quality of English LanguageModerate editing of English language still required.
Author Response
Reviewer 2:
“I would like to thank the authors for considering my comments. However, the modifications made including the changes in the Title, raise more questions. By replacing anti-inflammatory and pro-inflammatory with acute and chronic inflammatory, respectively, the authors failed to explain the biological criteria selected to define the two phases of inflammation. Knowing that acute inflammation causes the production of acute phase proteins, the authors must therefore provide the plasma concentration of C-reactive protein detected in the patients, for instance. In addition, IL-6, IL-1b, IL-8, TNF-a and TGF-b are well known as acute inflammatory interleukins, so the authors are strongly recommended to include at least IL-6 and IL-8 to the acute inflammatory IL signature.”
Point-by-point response to the reviewers’ comments
We want to thank the reviewer for taking the time to review this manuscript. We appreciate the reviewers’ comments, and we have prepared a revised version of the manuscript. Modifications in the manuscript are highlighted in green. Please find enclosed our detailed responses to the reviewers’ comments and suggestions.
Comment 1: “However, the modifications made including the changes in the Title, raise more questions. By replacing anti-inflammatory and pro-inflammatory with acute and chronic inflammatory, respectively, the authors failed to explain the biological criteria selected to define the two phases of inflammation.”
Response 1: Thank you for your helpful comment. The categorization of acute and chronic inflammatory signatures in our manuscript is based on their different roles within the tumor microenvironment (TME) rather than systemic inflammation. While acute inflammation in the broader sense is often associated with systemic markers such as C-reactive protein (CRP) and cytokines such as IL-6 and IL-8, our focus is limited to the localized immune responses within the breast cancer TME. The acute signature (including IL-12, IL-21, IFN-γ) and the chronic signature (including IL-4, IL-5, IL-10, IL-13, IL-17, CXCL1) modulate the TME differently: The acute inflammatory response in the TME creates a T helper (Th) type 1 profile, which promotes a tumor-suppressive immune milieu intended to destroy tumor cells. However, tumor immunoediting can lead to the development of tumor cell variants that evade this immune response, resulting in an equilibrium phase. As inflammation shifts from acute to chronic, a Th type 2 profile develops, which is associated with tumor-promoting effects, immune evasion, and uncontrolled tumor growth [1]. This transition underscores the distinct immune dynamics captured by the acute and chronic inflammatory signatures in our study.
To further clarify the differentiation between acute and chronic inflammation and its relevance within the tumor microenvironment in early breast cancer, we have included specific clarifications in the manuscript. These clarifications emphasize that the prognostic significance of the signatures we analyzed relates directly to localized inflammatory responses within the TME, rather than to systemic inflammation.
We have made the following modifications in the revised version of the manuscript:
- The present study aims to elucidate the prognostic impact of an acute inflammatory versus a chronic inflammatory interleukin mRNA expression signature within the TME of early breast cancer. (Lines 86-88)
- Overall, these results highlight the contrasting roles of specific interleukins within the TME in breast cancer prognosis. (lines 174/175)
- The acute inflammatory response, characterized by a Th1 profile, drives tumor suppressive immune activity in the TME, promoting the destruction of tumor cells. However, tumor cells can undergo immunoediting, allowing them to escape immune surveillance. As inflammation progresses from acute to chronic, a Th2 immune profile emerges, leading to immune evasion and tumor promotion [1]. This transition highlights the different immune dynamics captured by the acute and chronic inflammatory signatures in our study. (Lines 186-192)
The distinct prognostic roles of acute and chronic inflammatory interleukin signatures within the TME of breast cancer highlight the importance of a nuanced approach to immunotherapy. (Line 292)
Furthermore, the title reflects the TME-specific scope of our study and reflects the biological criteria underlying the definitions of acute and chronic inflammation within the TME in the manuscript.
Comment 2: “Knowing that acute inflammation causes the production of acute phase proteins, the authors must therefore provide the plasma concentration of C-reactive protein detected in the patients, for instance.”
Response 2: Thank you for your insightful comment regarding the role of C-reactive protein (CRP) as a marker of acute inflammation. As noted, CRP is indeed a key indicator of systemic inflammation, but our study focused on the tumor microenvironment rather than systemic responses. In this context, we reference findings from our own group (Sicking et al. 2014), where we observed no significant correlations between CRP levels in peripheral blood and the mRNA expression of key markers associated with inflammation (e.g. IL-6 and IL-8), immune response, or proliferation in breast cancer tissue [2]. This indicates that while CRP is an important systemic marker of inflammation, it does not necessarily reflect the localized immune processes within the TME.
In order to acknowledge your meaningful comment regarding the significance of CRP as a key marker of acute systemic inflammation and an important acute phase protein, we have added the following paragraph to the discussion.
“Interestingly, Sicking et al. evaluated the association between an acute systemic immune response (elevated serum CRP) and gene expression of immune markers of the TME at the mRNA level: they observed no significant correlations between peripheral blood CRP levels and mRNA expression of key markers associated with inflammation (e.g., IL-6 and IL-8), immune response or proliferation in breast cancer tissue. These findings suggest that although CRP is an important systemic marker of inflammation, it does not necessarily reflect localized immune processes within the TME.” (Lines 235-241)
Comment 3: “In addition, IL-6, IL-1b, IL-8, TNF-a and TGF-b are well known as acute inflammatory interleukins, so the authors are strongly recommended to include at least IL-6 and IL-8 to the acute inflammatory IL signature.”
Response 3: Thank you for your valuable suggestion regarding the inclusion of IL-6 and IL-8 in the acute inflammatory signature. We recognize that these cytokines are important mediators of inflammation, particularly as acute phase proteins in a systemic context. Within the breast cancer TME both IL-6 and IL-8 play critical roles in chronic inflammation and contribute to protumorigenic processes: IL-6 is involved in the activation of the STAT3 pathway, promoting tumor growth, immune evasion, and metastasis [3], while IL-8 enhances angiogenesis and neutrophil recruitment, further supporting tumor progression​ [4, 5]. Although IL-6 and IL-8 are key inflammatory cytokines, they are more characteristic of chronic inflammation and a protumorigenic microenvironment in cancer, rather than acute inflammatory responses. Therefore, their inclusion in the acute inflammatory signature would not accurately reflect the tumor-suppressive immune dynamics associated with this signature. However, to address your suggestion, we added IL-6 (205207_at) and IL-8 (202859_x_at, 211506_s_at) to the current acute inflammatory signature (in addition to IL-12, IL-21 and IFN-γ). The results showed that within the whole cohort, no significant effect on MFS was observed (Log Rank p= 0.720, Figure 1a). Similarly, the modified acute inflammatory signature did not show a significant effect on MFS using univariate (Table 1) and multivariate Cox regression analysis (Table 2). Considering the molecular subtypes, the prognostic effect of the modified acute inflammatory signature was limited to the luminal B subtype. Higher expression of the signature was associated with longer MFS (p=0.039, Log Rank, Figure 1b).
Overall, the inclusion of IL-6 and IL-8 in the acute inflammatory signature did not result in improved prognostic performance for metastasis-free survival. This further supports our decision to retain the current acute inflammatory signature (comprising IL -12, IL-21, IFN-γ) which more clearly represents the acute tumor suppressing Th1 based immune response in the breast cancer TME.
Table 1: Univariate Cox Regression analysis of the modified acute inflammatory signature (IL -12, IL-21, IFN-γ, IL-6, IL-8) in regard to MFS.
.
|
|
|
HR |
95% CI lower upper |
p- value |
|
|
Acute-inflammatory IL-Signature |
High vs. low |
0.940 |
0.669 |
1.321 |
0.721 |
Table 2: Multivariate Cox Regression analysis of the modified acute inflammatory signature (IL -12, IL-21, IFN-γ, IL-6, Il-8) in regard to MFS.
|
|
|
HR |
95% CI lower upper |
p- value |
|
|
Acute-inflammatory IL-Signature |
High vs. low |
0.768 |
0.499 |
1.181 |
0.229 |
|
Age |
<50 vs. ³50 |
1.112 |
0.644 |
1.922 |
0.703 |
|
Tumor size |
T2-4 vs. T1 |
1.552 |
0.975 |
2.471 |
0.064 |
|
Lymph node status |
N1,2,3 vs. N0 |
1.266 |
0.816 |
1.963 |
0.292 |
|
Grade |
GIII vs. GI/II |
2.253 |
1.401 |
3.623 |
<0.001 |
|
Ki-67 |
>20% vs. <20% |
1.324 |
1.401 |
3.623 |
0.220 |
- a)
b)
Figure 1 a) Kaplan Meier diagram of the modified acute inflammatory IL signature (IL-12, IL-21, IFN-, IL-6, IL-8) in the whole cohort (p=0.720 Log Rank). b) Kaplan Meier diagram of the modified acute inflammatory IL signature (IL-12, IL-21, IFN-, IL-6, IL-8) in the Luminal B subtype (p=0.039 Log Rank)
Finally, we would like to thank the reviewer for the valuable feedback that led to important revisions of the manuscript. We believe that the modifications in the revised version of the manuscript, including the biological clarification of the signatures and the detailed explanation of their roles in tumor suppression and promotion, have improved the quality and clarity of the study. We appreciate your careful consideration of the revised version and look forward to your feedback.
References:
- Schmidt, M. and A.S. Heimes, Immunomodulating Therapies in Breast Cancer-From Prognosis to Clinical Practice. Cancers (Basel), 2021. 13(19).
- Sicking, I., et al., Prognostic influence of pre-operative C-reactive protein in node-negative breast cancer patients. PLoS One, 2014. 9(10): p. e111306.
- Chen, J., et al., IL-6: The Link Between Inflammation, Immunity and Breast Cancer. Front Oncol, 2022. 12: p. 903800.
- Ahmed, S., et al., IL-8 secreted by tumor associated macrophages contribute to lapatinib resistance in HER2-positive locally advanced breast cancer via activation of Src/STAT3/ERK1/2-mediated EGFR signaling. Biochim Biophys Acta Mol Cell Res, 2021. 1868(6): p. 118995.
- Habanjar, O., et al., Crosstalk of Inflammatory Cytokines within the Breast Tumor Microenvironment. Int J Mol Sci, 2023. 24(4).